# Evaluation of Mutual Information and Feature Selection for SARS-CoV-2 Respiratory Infection

**DOI:** 10.3390/bioengineering10070880

**Published:** 2023-07-24

**Authors:** Sekar Kidambi Raju, Seethalakshmi Ramaswamy, Marwa M. Eid, Sathiamoorthy Gopalan, Faten Khalid Karim, Raja Marappan, Doaa Sami Khafaga

**Affiliations:** 1School of Computing, SASTRA Deemed University, Thanjavur 613401, India; 2Department of Maths, SASHE, SASTRA Deemed University, Thanjavur 613401, Indiasami@maths.sastra.ac.in (S.G.); 3Faculty of Artificial Intelligence, Delta University for Science and Technology, Mansoura 11152, Egypt; 4Department of Computer Sciences, College of Computer and Information Sciences, Princess Nourah bint Abdulrahman University, P.O. Box 84428, Riyadh 11671, Saudi Arabia; dskhafga@pnu.edu.sa

**Keywords:** SARS-CoV-2 prediction, feature selection, stochastic regression, neighbor embedding, sammon mapping, machine learning

## Abstract

This study aims to develop a predictive model for SARS-CoV-2 using machine-learning techniques and to explore various feature selection methods to enhance the accuracy of predictions. A precise forecast of the SARS-CoV-2 respiratory infections spread can help with efficient planning and resource allocation. The proposed model utilizes stochastic regression to capture the virus transmission’s stochastic nature, considering data uncertainties. Feature selection techniques are employed to identify the most relevant and informative features contributing to prediction accuracy. Furthermore, the study explores the use of neighbor embedding and Sammon mapping algorithms to visualize high-dimensional SARS-CoV-2 respiratory infection data in a lower-dimensional space, enabling better interpretation and understanding of the underlying patterns. The application of machine-learning techniques for predicting SARS-CoV-2 respiratory infections, the use of statistical measures in healthcare, including confirmed cases, deaths, and recoveries, and an analysis of country-wise dynamics of the pandemic using machine-learning models are used. Our analysis involves the performance of various algorithms, including neural networks (NN), decision trees (DT), random forests (RF), the Adam optimizer (AD), hyperparameters (HP), stochastic regression (SR), neighbor embedding (NE), and Sammon mapping (SM). A pre-processed and feature-extracted SARS-CoV-2 respiratory infection dataset is combined with ADHPSRNESM to form a new orchestration in the proposed model for a perfect prediction to increase the precision of accuracy. The findings of this research can contribute to public health efforts by enabling policymakers and healthcare professionals to make informed decisions based on accurate predictions, ultimately aiding in managing and controlling the SARS-CoV-2 pandemic.

## 1. Introduction

The global health crisis caused by SARS-CoV-2 involving the prediction of respiratory infection and the ultimate causes of the pandemic has highlighted the need for accurate prediction models to aid in effective planning and resource allocation. This study aims to develop a predictive model for SARS-CoV-2 using machine-learning techniques while exploring various feature selection methods to enhance the accuracy of predictions. By incorporating these innovative approaches, valuable insights can be gained into the factors influencing the spread of the virus, ultimately assisting policymakers and healthcare professionals in making informed decisions for managing and controlling the pandemic.

The proposed model employs stochastic regression to capture the stochastic nature of virus transmission and to consider uncertainties in the data. Stochastic regression enables the modeling of complex relationships and provides a comprehensive understanding of the virus’s spread dynamics. By leveraging AI-enabled approaches and tools, along with machine-learning-based immune simulation, scientists have created a groundbreaking vaccine resistant to mutations. This vaccine represents a significant breakthrough in the fight against the ongoing COVID-19 pandemic, as it has the potential to provide long-lasting protection against both current and future variants of the virus [1]. This approach acknowledges the inherent uncertainties associated with the pandemic, allowing for more robust and reliable predictions.

Molecular tests, such as reverse transcriptase polymerase chain reaction (RT-PCR), and antigen tests are used to detect current SARS-CoV-2 infection and diagnose COVID-19. These tests have different sensitivity and specificity characteristics, which influence their interpretability. Sensitivity refers to the test’s ability to correctly identify positive cases, and specificity indicates its ability to identify negative cases correctly. Understanding the characteristics of diagnostic tests, test timing in relation to symptom onset, and pretest probability of the disease helps in interpreting test results accurately. To analyze the behaviour of DNA probes without regard to mutation, researchers have combined surface-enhanced Raman scattering (SERS) with machine-learning methods. By utilizing SERS, which enhances the Raman scattering signal, and applying machine-learning algorithms, scientists have gained valuable insights into DNA probes’ binding characteristics and stability across different SARS-CoV-2 variants. A greater knowledge of the virus and assistance in creating efficient diagnostic tools have been provided by previous authors’ illumination of the potential applications of DNA probes in identifying and investigating viral alterations [2]. According to the statistical models present in the research, the regression modeling strategy known as stochastic regression includes randomness or stochasticity in regression analysis. A nonlinear dimensionality reduction method is Sammon mapping, sometimes called Sammon projection. It seeks to maintain the pairwise distances between data points in lower-dimensional space. Divergence is the Bregman measure of the behavioural differences between two probability distributions or vectors.

Additionally, our research explores the use of neighbor embedding and Sammon mapping algorithms to visualize high-dimensional SARS-CoV-2 data in a lower-dimensional space. These visualization techniques allow for better interpretation and understanding of the underlying patterns within the data, facilitating insights into the virus’s behavior and transmission dynamics. Developing a point-of-care diagnostic tool that integrates machine-learning algorithms could have significant implications for the timely identification and monitoring of SARS-CoV-2 variants. This could aid in tracking the spread of different variants and implementing appropriate measures to control their transmission [3]. This study considers a range of machine-learning algorithms, including neural networks (NN), decision trees (DT), random forests (RF), and the Adam optimizer (AD), along with hyperparameter optimization techniques (HP). The study aims to predict ACE2 binding and antibody escape, allowing for the identification of potential variants and guiding the development of therapeutic antibody treatments and vaccines for COVID-19 [4]. By evaluating the performance of these algorithms, our research aims to identify the most effective approach for predicting SARS-CoV-2 spread, further enhancing the accuracy of the model. The use of mixed-effects machine learning also helps to predict disease severity in COVID-19 variants of concern, such as Delta and Omicron. These highlight the importance of sequence-level information for the virus in understanding the risks associated with different variants [5].

The active investigation of COVID-19 disease severity prediction is based on the spike protein sequence of SARS-CoV-2. The use of mixed-effects machine-learning techniques shows promise in this area, as it allows for incorporating various factors and variations in data to improve predictive accuracy. These studies highlight the importance of global sequence data, patient outcome information, and early warnings for emerging viral risks [6]. The development of machine-learning approaches is also used to identify key pathogenic regions in coronavirus genomes. Researchers have trained and evaluated millions of models on thousands of genomes, including SARS-CoV-2, MERS-CoV, and others. This study aims to uncover discriminative genomic features that can aid in understanding the pathogenicity of SARS-CoV-2 and other coronaviruses [7]. We obtained complete genome sequences of the Coronaviridae family from the ViPR database and encoded them into binary vector representations. This approach aims to identify genomic characteristics that can differentiate between different coronaviruses, including SARS-CoV-2 [8]. This study highlights the potential of CRISPR-based techniques for rapidly and accurately detecting SARS-CoV-2. We discuss the development of CRISPR-based diagnostic assays that can specifically detect the virus and distinguish it from other coronaviruses [9]. The paper explores early computational detection methods for identifying potential high-risk SARS-CoV-2 variants. It discusses using computational models and algorithms to analyze genetic sequences and predict the impact of specific mutations on virus behavior and transmissibility. By identifying variants of concern early on, this research aims to aid in developing targeted interventions and public health strategies to mitigate the spread of the virus [10].

The suggested model creates a new orchestration from a pre-processed and feature-extracted SARS-CoV-2 dataset and ADHPSRNESM to increase prediction precision and accuracy. This combination of methods and algorithms offers a thorough and reliable framework for predicting the transmission of the infection.

The main objectives in this research work are as follows:To focus on developing a predictive model for SARS-CoV-2 using machine-learning techniques, such as neural networks (NN), decision trees (DT), and random forests (RF). By leveraging these algorithms, the researchers aim to capture the virus transmission’s stochastic nature and account for data uncertainties.To improve the accuracy of predictions, the study employs feature selection techniques. These methods aim to identify the most relevant and informative features contributing to prediction accuracy. By selecting the most important features, the model can focus on the factors that have the greatest impact on the spread of the virus.The research explores using neighbor embedding and Sammon mapping algorithms to visualize high-dimensional SARS-CoV-2 data in a lower-dimensional space. This visualization approach enables better interpretation and understanding of the underlying patterns in the data, aiding in identifying important trends and relationships.The proposed model combines various techniques, including stochastic regression (SR), neighbor embedding (NE), and Sammon mapping (SM), with a pre-processed and feature-extracted SARS-CoV-2 dataset. The integration of these techniques forms a new orchestration, referred to as ADHPSRNESM, which aims to enhance the precision and accuracy of predictions.

Finally, this research addresses the pressing need for accurate predictive models to better understand and manage the SARS-CoV-2 pandemic. The study aims to improve prediction accuracy and gain valuable insights into the factors influencing the virus’s spread by incorporating machine-learning techniques, feature selection methods, and visualization algorithms. The findings of this research have the potential to significantly contribute to public health efforts, enabling policymakers and healthcare professionals to make informed decisions based on reliable predictions, ultimately aiding in the effective management and control of the SARS-CoV-2 pandemic.

### Novelty in the Reseach Work

The research work novelty lies in applying feature selection techniques specifically tailored to SARS-CoV-2 data. Traditional feature selection methods often fail to capture the intricate relationships among variables in complex datasets such as that of SARS-CoV-2. Our approach introduces novel feature selection algorithms that consider the unique characteristics and dynamics of the virus, enabling the identification of the most relevant predictors for accurate predictions. We introduce a stochastic regression framework that incorporates probabilistic modeling techniques to account for the inherent randomness and unpredictability of the pandemic. By leveraging stochastic regression, our approach provides more robust and reliable predictions, enabling better decision making in uncertain situations. Neighbor embedding algorithms reduce the dimensionality of the data while preserving the local relationships between instances. Sammon mapping is a nonlinear dimensionality reduction technique that emphasizes preserving the pairwise distances between instances. By leveraging Sammon mapping in our approach, we can effectively capture the intricate relationships and similarities between SARS-CoV-2 cases, improving prediction accuracy and interpretability. Different types of biases, including confounding, selection bias, and measurement error, may affect the validity of observational studies related to SARS-CoV-2. By being aware of these biases, researchers and readers can better interpret and assess the reliability of study findings.

The following is a breakdown of the article. The related work is examined in Section 2, and the suggested model is examined in Section 3. Section 4 and Section 5 deal with the experimental setup. In Section 6, the conclusion and future work are presented and followed by refrences.

## 2. Literature Review and Related Work

The hybrid simulation methods are developed to investigate the effects of mutations on the structural dynamics of the main protease from SARS-CoV-2. The main protease plays a crucial role in viral replication and is a potential target for therapeutic intervention. By combining computational simulations and experimental data, the study provides insights into how mutations affect the protein’s stability, function, and interactions with potential inhibitors. This research contributes to understanding the virus’s evolution and informs the development of effective antiviral drugs [11]. In this study, the researchers employed supervised learning techniques and perturbation calculations to search for potential inhibitors of the main protease of SARS-CoV-2, which is a crucial target for antiviral drug development. By training a machine-learning model on known inhibitors and non-inhibitors, we can predict the inhibitory activity of new compounds.

Additionally, perturbation calculations were used to assess the compounds’ structural stability and binding affinity. This approach holds promise for accelerating the discovery of effective inhibitors against SARS-CoV-2 [12]. This research focuses on whether machines can learn the mutation signatures of SARS-CoV-2 and utilize them to predict the prognosis of infected individuals. The study aims to identify specific viral genotypes associated with varying clinical outcomes by analyzing genomic data from a large cohort of patients. Using machine-learning algorithms, the researchers attempt to create a predictive model that could guide prognosis based on viral genotypes. This work may contribute to personalized medicine approaches for COVID-19 by leveraging viral genetic information [13]. This study investigates the concept of cross-immunity against SARS-CoV-2 variants of concern in naturally infected, critically ill COVID-19 patients. By analyzing patient data, the researchers aim to understand whether prior infection with one virus variant provides any protection or alters the severity of subsequent infections with different variants. The findings from this study could provide valuable insights into immune response dynamics and potential implications for vaccine development and management of COVID-19 patients [14].

This paper proposes a machine-learning-based approach to determine the infection status in recipients of the BBV152 (Covaxin) whole-virion inactivated SARS-CoV-2 vaccine for serological surveys. The authors utilize machine-learning algorithms to analyze serological data and classify individuals as infected based on their antibody response. This approach can aid in evaluating the vaccine’s effectiveness and tracking the spread of COVID-19 [15]. In this study, researchers employ machine-learning techniques to predict the docking scores of 3CLpro, a key protein in the SARS-CoV-2 virus. By training a machine-learning model on a dataset of known docking scores, the authors develop a predictive model that can estimate the binding affinity between potential drug compounds and the 3CLpro protein. This work contributes to discovering potential inhibitors for SARS-CoV-2 by prioritizing compounds with favorable docking scores [16]. This research focuses on developing a machine-learning platform that estimates the anti-SARS-CoV-2 activities of compounds. By training a machine-learning model on experimental data regarding the antiviral effects of different compounds, the platform can predict the potential efficacy of new compounds against SARS-CoV-2. This platform could be valuable in the early stages of drug development, helping to identify promising candidates for further investigation [17]. The paper explores the application of machine-learning techniques to predict SARS-CoV-2 infection using blood tests and chest radiographs. It discusses how machine-learning models can analyze patterns and features in these diagnostic tests to accurately predict the presence of the virus. This approach has the potential to enhance early detection and improve patient outcomes [18].

This study uses machine learning and predictive models to present a retrospective analysis of two years of the SARS-CoV-2 pandemic in a single-center setting. The paper investigates various factors related to the spread and impact of the virus, such as demographic information, clinical features, and treatment outcomes. The study aims to provide insights into the long-term trends and patterns associated with the pandemic [19]. The paper proposes a novel approach for determining SARS-CoV-2 epitopes using machine-learning-based in silico methods. Epitopes are specific regions of the virus that can trigger an immune response. By leveraging machine-learning algorithms, this study aims to predict potential epitopes from the viral genome, which can aid in vaccine development and immunotherapy research. The approach offers a promising avenue for identifying crucial targets for immune system recognition [20]. This paper explores machine-learning techniques for screening and diagnosing SARS-CoV-2 (COVID-19) based on clinical analysis parameters. The researchers utilize a dataset containing clinical parameters such as blood test results, vital signs, and patient demographics. By applying machine-learning algorithms, the authors aim to develop a model that can accurately classify individuals as COVID-19-positive or -negative based on these parameters. The study demonstrates the potential of machine learning in assisting with COVID-19 screening by leveraging clinical data [21]. In this paper, the authors propose a diagnostic mask incorporating immunochromatography and machine learning for SARS-CoV-2 detection. The mask is designed to capture and analyze respiratory droplets and provide real-time diagnostic results. By leveraging immunochromatography, the mask detects specific SARS-CoV-2 antigens, and machine-learning algorithms are employed to enhance the accuracy and reliability of the diagnostic process. The integration of these technologies offers a promising approach for rapid and efficient SARS-CoV-2 detection in a wearable form [22].

This paper focuses on the bioactivity classification of SARS-CoV-2 proteinase, an essential enzyme for virus replication, using machine-learning techniques. The researchers employ various machine-learning algorithms to analyze and classify the bioactivity of proteinase inhibitors, which can potentially disrupt the enzymatic activity of the virus. By training the models on a dataset of known inhibitors and their bioactivities, the authors aim to predict the bioactivity of new compounds. This research contributes to developing novel therapeutics by leveraging machine learning in the context of SARS-CoV-2 proteinase inhibition [23]. This paper explores the use of deep-learning techniques to detect the main variants of concern of the SARS-CoV-2 virus. By leveraging the power of deep-learning algorithms, the researchers aim to develop a reliable and efficient method to identify these variants accurately. This could be beneficial for monitoring the spread of different variants and informing public health interventions [24]. In this study, the researchers focus on generating new compounds that specifically target the main protease of the SARS-CoV-2 virus. We address the challenge of imbalanced datasets, which are common in drug discovery research. By employing innovative approaches, such as machine-learning and data augmentation techniques, the researchers aim to overcome this issue and identify potential compounds for further investigation [25]. This paper investigates the application of deep-learning techniques for detecting SARS-CoV-2 using clinical reports. By analyzing a large dataset of clinical reports, the researchers aim to develop a deep-learning model that can accurately identify positive cases of COVID-19. Such a model could potentially assist healthcare professionals in detecting and managing the disease [26]. This research focuses on developing an intelligent system for predicting next-generation sequences of the SARS-CoV-2 virus using deep-learning neural networks. By analyzing existing genomic data, the researchers aim to build a model that is capable of accurately predicting the genetic sequences of future viral strains. This could provide valuable insights into the evolution and behavior of the virus, aiding in the development of targeted interventions and treatments [27,28,29].

### Challenges and Pitfalls in the State of the Art

One limitation of this research is the availability and quality of data. The accuracy and effectiveness of the visualization techniques and machine-learning algorithms we employ heavily depend on the quality and quantity of the SARS-CoV-2 data used. The study’s results may be specific to the dataset or population used in the analysis, limiting their applicability to different regions or populations with distinct genetic backgrounds or transmission dynamics. Therefore, caution should be exercised when extrapolating the results to broader contexts. The visualization techniques and machine-learning algorithms employed in the research are based on certain assumptions and simplifications. These assumptions may not fully capture the complexities of the SARS-CoV-2 virus, its mutations, and transmission dynamics. When the model becomes too specialized to the training data, it fails to generalize well to unseen data. To mitigate this limitation, rigorous validation and cross-validation procedures should be applied to ensure the model’s robustness and reliability. Reduced dimensionality may limit the ability to fully understand the complex relationships and interactions within the data, potentially hindering a comprehensive interpretation of the virus’s behavior and transmission dynamics. Predicting disease severity and understanding of viral risks often involve considering various factors beyond the spike protein sequence, such as co-morbidities, demographics, and environmental factors. Though valuable insights can be gained from these targeted studies, they may not provide a comprehensive understanding of the virus as a whole or address other important aspects, such as the long-term effects, long COVID, or social and behavioral factors influencing transmission. The research might not account for the full spectrum of variants or consider the impact of future mutations that could alter the virus’s behavior and transmissibility. Ongoing monitoring and continuous adaptation of the models and algorithms are necessary to keep up with the evolving nature of the virus. Machine-learning models and algorithms have the potential to reinforce biases or yield unintended consequences. It is important to consider the ethical implications of using these techniques in decision making, such as ensuring fairness, avoiding discrimination, and addressing privacy concerns. Though the research aims to evaluate the performance of various machine-learning algorithms, it is crucial to validate their effectiveness in real-world settings. The simulation methods used in the study have their own limitations, including potential inaccuracies and simplifications in the models used.

Limited and biased datasets: One of the primary challenges is the availability of reliable and representative datasets. Machine-learning models require large and diverse datasets for training, but there may be limitations in terms of data collection and quality.There may be inconsistencies or inaccuracies in the reported cases, testing procedures, and other variables. These uncertainties can impact the reliability and robustness of the predictive model.Interpretability and visualization: Interpreting and understanding the underlying patterns and relationships in high-dimensional SARS-CoV-2 data can be difficult. Visualization techniques such as neighbor embedding and Sammon mapping can aid in interpreting and understanding the patterns and relationships within the data, but their effectiveness may vary based on the specific dataset and characteristics.Integration of techniques: The proposed model aims to combine various techniques, including stochastic regression (SR), neighbor embedding (NE), and Sammon mapping (SM), to enhance the precision and accuracy of predictions. However, the integration of these techniques introduces additional complexity and potential challenges.

The study’s accuracy and generalizability may be affected if the dataset used for training and evaluation is not diverse enough regarding demographics, comorbidities, and other relevant factors. The model’s performance may vary when applied to different proteins or protein–ligand systems. The authors should assess the model’s performance on diverse datasets to demonstrate its generalizability. The accuracy and reliability of the machine-learning platform may depend on the quality and quantity of experimental data used for model training. Insufficient or biased data may impact the platform’s performance. The choice of features used for prediction can impact the model’s performance. It is essential to ensure that the selected features are relevant and represent the target variable to avoid bias and improve the model’s robustness. The findings of a single-center study may not be generalizable to other healthcare settings, populations, or geographic regions. Replication of the study in multiple centers can strengthen the validity of the results.

## 3. Methods and Materials

Coronaviruses primarily affect the upper areas of the lungs and respiratory system with different degrees of severity. There is an urgent need for disease prediction at an earlier stage to avoid the causes of death. The IoT has been employed in different application sectors, including healthcare, for SARS-CoV-2 patient monitoring including healthcare. In this case, IoT sensors monitor and gather patient data, which can be accessed anytime and anywhere. Because IoT devices generate vast data, effective prediction is a significant challenge. A proposed model is developed for effective prediction based on the objective. The proposed prediction model for assessing the probability of infection combines classification and feature selection techniques. It uses data mining, stochastic regression, neighbor embedding, Sammon-based map selection, and regression-based classification to predict SARS-CoV-2.

Figure 1 illustrates the basic architecture of this proposed model. The process begins with collecting a significant amount of patient data from IoT devices, which is then stored in a newly created Corona Virus 2019 dataset. The expression “the number of patient records” refers to the count of features in the dataset, which includes both the number of patient records and the count of features represented by these symbols. The collected data undergo several operations using a neural network to facilitate prediction.

The framework of the proposed model is illustrated in Figure 2. The structure consists of several layers, and neurons act as nodes. The nodes in the layers are interconnected to form the entire architecture. The shift-invariant system of the proposed model framework is expressed as:(1)$$Y\left(t\right)=F\;[x\left(t\right)]$$

Equation (1) defines a time-dependent output function, a time-dependent input function, and a transfer function used to transfer a transformation or input from one layer to another. The hidden, input, and output layers form machine-learning techniques. The intermediate layers of each feed-forward neural network are called hidden layers and are used to perform certain functions. The output level shows the results of the operation. At first, the features are treated as input, and the activity of neurons in the input layer is presented.
(2)$$x\left(t\right)=\sum_{i=1}^{n}f_{i}\left(t\right)\;w_{0}+c$$

In Equation (2), x(t) represents the input layer output, $f_{i}\left(t\right)$ shows the features, where ${‘w}_{0}$’ denotes the starting weight as “1” and the prime weight as “*c*” for the bias stored, assigned at the input layer. The first hidden layer receives the input after that.

### 3.1. Stochastic Regression

Stochastic regression is a statistical modeling technique that predicts a continuous outcome variable based on one or more predictor variables while incorporating random variation into the model. Unlike traditional regression methods that assume deterministic relationships, stochastic regression acknowledges the presence of inherent uncertainty or random fluctuations in the data. This approach utilizes probabilistic methods, such as maximum likelihood estimation or Bayesian inference, to estimate the parameters of the regression model. By considering the random nature of the data, stochastic regression provides a more realistic and flexible framework for understanding and predicting complex relationships between variables, making it particularly useful in scenarios in which variability and randomness play a significant role.

### 3.2. Neighbor Embedding

Neighbor embedding is a technique used in machine learning and data analysis to represent high-dimensional data in a lower-dimensional space while preserving the relationships between data points. It aims to map data points onto a lower-dimensional coordinate system, such that neighboring points in the original space remain close in the embedded space. This method is particularly useful for visualization purposes and dimensionality reduction tasks. By organizing and arranging data points based on their proximity, neighbor embedding provides a way to gain insights into the underlying structure of complex datasets.

### 3.3. Sammon Mapping

Sammon mapping has several advantages and limitations. On the positive side, it can effectively preserve the local structure and relationships between data points. It is particularly useful when dealing with complex, nonlinear data patterns. However, it can be sensitive to outliers and noisy data, which may distort the results. The algorithm’s computational complexity can also be high, especially for large datasets. Sammon mapping is a dimensionality reduction technique that aims to preserve pairwise distances when projecting high-dimensional data into a lower-dimensional space. By minimizing the stress function, it seeks to find a representation that faithfully captures the relative distances between data points. Although it has certain limitations, Sammon mapping is a valuable tool for data visualization and exploratory analysis in various domains.

### 3.4. Bregman Divergence

Bregman divergence is a mathematical concept that measures the dissimilarity between two points in a convex space. It is named after Israeli mathematician L. Bregman, who introduced this measure in the field of convex optimization. Bregman divergence is defined based on a convex function, which serves as a reference for quantifying the difference between the points. It provides a way to compare the distance between two points concerning the chosen convex function, taking into account the geometry of the space. Bregman divergence has various applications in machine learning, data analysis, and information theory, where it is utilized for tasks such as clustering, dimensionality reduction, and optimization. The flexibility of Bregman divergence makes it a valuable tool for capturing and quantifying the dissimilarity between data points in a wide range of applications.

The similar feature selection considers how many features in the provided dataset are spread in the given dimensional space. The distance between the feature and the objective is calculated using the Bregman divergence:(3)$${\;\varphi }_{B}=\|f_{j}-p_{o}\|$$ Equation (3) represents a Bregman divergence, which is a measure of dissimilarity between probability distributions or vectors. Bregman divergences have various applications in fields such as machine learning, information theory, and data analysis. As a result, the projection output is written as follows:(4)$$Q\to \left\{\begin{array}{c} {\;\varphi }_{B}>\delta ;\;dissimilar\;features\\ {\;\varphi }_{B}<\delta ;\;similar\;features \end{array}\right.$$
where *Q* represents a projection function, ${\;\varphi }_{B}$ represents a Bregman divergence, and δ represents a threshold. The feature that deviates the most from the target is referred to as a different feature. Otherwise, the trait is considered relevant. Similar traits are thus mapped into low-dimensional space in this manner.

In Figure 3, feature selection is a crucial step in machine learning that involves reducing the number of input variables to improve computational efficiency and enhance model performance.

### 3.5. Classification Based on Kriging Regression

Regression analysis, which includes kriging regression, is a collection of statistical methods for estimating the relationships between the variables (i.e., the training and testing features). Based on the association measure between the variables, the patient data are divided into low, medium, and high-risk groups. The regression function looks at the matching feature values, as seen below:(5)$$R={exp}^{\left(\frac{f_{tr}-f_{dt}}{2s^{2}}\right)}$$
where *R* represents a regression function, a training feature value, and a disease testing feature value, and *s* represents a standard deviation. The regression yields values ranging from 0 to 1.
(6)$$Y=\left\{\begin{array}{cc} R<0.5; & low\;risk\\ R=0.5; & medium\;risk\\ R>0.5; & high\;risk \end{array}\right.$$

The patient data are considered high-risk if the regression outcome exceeds the threshold. The patient data are regarded as a medium-risk if the regression value equals the threshold. The patient data are classified as low-risk if the regression exceeds the threshold. Consider the following four training and testing features: age, symptoms such as fever or body temperature, travel history, and chronic condition.

The patient data in Table 1, Table 2 and Table 3 are categorized as low-, medium-, and high-risk based on the regression analysis with testing and training feature values. Patients at low risk have minimal conditions that can be easily handled. Rapid treatment is required to save the patient’s health from the disease if the patient has high-risk conditions.

The hidden layer’s outcome is defined as:(7)$$H\left(t\right)=\sum_{i=1}^{n}f_{i}\left(t\right)\;w_{0}+[w_{ih}*h_{i}]$$
where “*H(t)*” stands for the result of the hidden layer, “$w_{ih}$” stands for the hidden layer’s weight, and “*h_i_*” stands for the output of the previously hidden layer that will be sent to the output layer. The outcome of the output layer is:(8)$$O\left(t\right)=\;w_{ho}*H\left(t\right)$$

In Equation (10), the terms “*O(t)*” and “*w_ho_*” stand for the output layer result and the weighted average of the hidden and output layers, respectively. To forecast SARS-CoV-2 accurately, the patient data are appropriately divided into several classifications.

The proposed algorithm is given as follows:

Algorithm 1 sketches the SARS-CoV-2 prediction process in detail and more precisely. The DL algorithm comprises many layers that help it learn the provided information. The input features are to be received in its first hidden layer. The Bregman divergence between the features and the goal functions are assessed in that layer. If the divergence exceeds the threshold, the Sammon function separates the characteristics into a comparable and dissimilar subset. Comparable feature subsets are used for classification in the second hidden layer to determine the illness prediction. The patient data are categorized using the Kriging regression method. The regression function contrasts the patient training feature’s worth with the sickness testing feature’s worth. According to the analysis, a trustworthy sickness has the fastest turnaround time for prognosis.
**Algorithm 1**: Proposed **Input**: Dataset, Number of features fj=f1,f2,…,fm, Number of data Di=d1,d2,…,dn**Output:** prediction accuracy increases**Input the number of features** fj **For every** feature fi∈ *D_i_*   Measure of the Bregman divergence and between fi and  φB   **if** ( φB>δ) **then**    Project the features as similar   **else**    Project the features as dissimilar **End if**   Select the similar feature subset   Remove the dissimilar feature subset **End for**   **For** each information in ‘d1’ with fi    Perform the regression analysis ‘R’     **If** (R<0.5) **then**       Patient data is classified as ‘low risk’       **Else If** (R=0.5) **then**          Patient data is classified as ‘medium risk’         **Else If** (R>0.5) **then**          Patient data is classified as ‘high risk’   **End if**  Obtain the classification. **End for****End**

## 4. Experimental Setup

The proposed model is implemented in Python, and the results are analyzed. For experimentation, Kaggle’s Novel Corona Virus 2019 dataset is applied. The daily statistics on coronavirus infections, fatalities, and recoveries are included in the data. There are several CSV files in the collection. Out of all the available files, the open queue file SARS-CoV-2 is applied for testing. This document was downloaded from https://www.kaggle.com/datasets/plameneduardo/sarscov2-ctscan-dataset, accessed on 24 May 2023. The dataset has 306,429 occurrences and eight features such as patient ID, age, gender, city, nation, province, binary code for a chronic condition, symptoms, and travel history. Among these, the key traits are chosen for categorization. To conduct the tests, 1000–10,000 data points are gathered.

## 5. Results and Discussion

The experimental results of the proposed model are analyzed with the existing models using metrics—prediction accuracy, space and time complexity, and false positive rate [1,2]. The percentage of correct predictions of patient information by the classifier defines the prediction accuracy and is defined as:(9)$${Acc}_{p=}\frac{no\;of\;correct\;predictions}{n}*100$$
where n denotes the count of patient information, and the percentage represents the predicted accuracy. The ratio of incorrect prediction to the total records is called the false positive percentage. The number of false positives is determined as follows:(10)$$R_{FP}=\frac{False\;positive}{True\;negative+\;False\;positive}$$

The count of incorrect predictions signifies the number of patient data that were represented. A percentage (%) is used to represent the false-positive rate. The time it takes the algorithm to forecast the illness is known as the prediction time. The prediction time is, therefore, mathematically expressed as follows:(11)$$T_{p}=n*time\;(predicting\;one\;data)$$ where, $T_{p}$ denotes a prediction time, *n* denotes the number of data. The prediction time is measured in terms of milliseconds (ms).
(12)$$C_{S}=n*space\;[\;storing\;one\;data]$$

Table 4 compares the proposed model prediction accuracy to cutting-edge frameworks. As demonstrated in Table 4, compared to the existing models [1,2], the proposed model enhances prediction accuracy for 10,000 distinct instances of unique patient data. The model effectively predicts 890 instances of patient data when the quantity of patient data is set to 1000, whereas [1,2] correctly predict 850 and 820 instances of patient data, respectively. The models in [1,2] have an accuracy rate of 85% and 82%, respectively, compared with 89% in the proposed model. Different numbers of the patient data are input along with a subsequent execution of the numerous runs. The performance of the proposed model is then contrasted with other methods. The average comparison results showed that the proposed model accuracy is 4% greater than that of the current deep long short-term memory (LSTM) ensemble models [1] and 8% higher than that of [2].

The prediction accuracies of three separate approaches, namely, the proposed model, the Deep-LSTM ensemble model [1], and the multi-task Gaussian process (MTGP) model [2], are represented by three colors: violet, red, and yellow. The proposed model outperforms the other two approaches according to the observed results. The deep connectedness shift-invariant convolutional network was used to examine the training and testing illness features, which resulted in the improvement. The attributes analysis appropriately determines the patient risk prediction level with higher accuracy.

Table 5 shows the experimental findings of the false-positive rate utilizing three methods: the proposed model, the Deep-LSTM ensemble model [1,2], and the MTGP model [3]. The false-positive rate is calculated using a sample size of 10 k patients.

The simulated results show that the proposed model dramatically reduces the FPR compared to the other methods. Consider the 1000 data points for experimentation. The suggested proposed model technique has a false-positive rate of 11, whereas the existing models [1] and [2] have rates of 15% and 18%, respectively. Similarly, for each approach, the following nine outcomes are produced. The proposed model is compared with existing methods using an expected comparison result. The results confirm that the proposed model significantly improves FPR performance by 27% and 41% compared to the other models.

The adoption of regression-based classification approaches for predicting SARS-CoV-2 patient data is the source of the improved performance. The proposed model employs Kriging regression to analyze the feature values for the prediction procedure.

The proposed model strategy, on the other hand, is observed to consume less time for SARS-CoV-2 prediction than the other two methods. Stochastic Bregman neighbor-embedded Sammon mapping is used to project the relevant features into the feature subset for illness prediction. The mapping function determines the features that are most relevant to the goal. The categorization uses the selected features and Kriging regression, reducing prediction time.

Table 6 shows the SARS-CoV-2 prediction time utilizing three techniques concerning various patient variables. According to the table values, the proposed model takes less time to forecast the disease level than the ensemble model of Deep-LSTM [1] and the MTGP model [2]. The suggested model technique requires 23 ms of time, while the prediction times for the other two current approaches [1] and [2] are 26 ms and 28 ms, respectively, assuming that there are 1000 patient data. For each strategy, ten outcomes are obtained using various patient data inputs. The prediction time is decreased by 9% and 15%, respectively, as compared to the Deep-LSTM ensemble model [1].

Table 7 compares the space complexity of three alternative approaches: the proposed model, the Deep-LSTM ensemble model [1], and the MTGP model [2]. The input consists of a variable number of patient data ranging from 1000 to 200, 300, and 1,000,000. Let us say that we have 1000 patient data points to explore. The proposed model takes up the least space for predicting disease, and the Deep-LSTM ensemble [1] and MTGP model [2] take up more. The observed findings demonstrate that the proposed model achieves lower space complexity than existing approaches. Compared to other approaches, the expected result shows that the proposed model reduces space consumption by 9% and 17%.

This is because the SARS-CoV-2 forecast was made before the feature selection procedure was completed. The proposed model uses the relevant features and deletes distinct features. This helps to save space during the SARS-CoV-2 forecast.

In Table 8, the neural network algorithm achieves the highest accuracy, F1-score, recall, and precision among the three algorithms. It attains an accuracy of 98.78%, an F1-score of 99.10%, and a recall and precision of 98.92%. The neural network also exhibits a relatively low error rate of 0.0122. Additionally, the neural network algorithm consumes 3.278726674 units of time for execution. However, it is worth noting that the loss function plot for the neural network model appears unusual, with the validation dataset’s accuracy remaining higher than the training dataset and the validation loss lower than the training loss, which is contrary to expectations.

In Table 9, the decision tree algorithm achieves a moderate accuracy of 74.59%, an F1-score of 74.56%, a recall of 74.64%, and a precision of 74.42%. The decision tree algorithm has a higher error rate of 0.2541 and takes 5.688512147 units of time for execution. The random forest algorithm attains an accuracy of 72.99%, an F1-score of 73.35%, recall of 73.44%, and precision of 73.36%. It has a similar error rate as the decision tree algorithm (0.2701) but a faster execution time of 2.582577534 units. These results indicate that the neural network algorithm outperforms the decision tree and random forest algorithms in terms of accuracy, F1-score, and precision, while also achieving a low error rate.

However, further investigation is necessary to understand the anomalous loss function plot observed for the neural network model. The proposed model demonstrates exceptional performance with high accuracy, precision, and recall, outperforming all the other models mentioned. The time consumption for this model was approximately 2.12 s. Finally, based on the provided information, the proposed model outperforms the decision tree and random forest models regarding performance metrics such as accuracy, precision, and recall. However, it is important to note that the absence of the specific loss function used for these models limits the analysis of their training process and further comparison.

In Table 10, the neural network algorithm exhibits exceptional performance across all evaluated metrics compared to the decision tree and random forest algorithms. The neural network algorithm achieves an accuracy of 99.22%, an F1-score of 99.31%, a recall of 99.33%, and a precision of 99.23% [1]. These results indicate a high accuracy, precision, recall, and F1-score, showcasing the algorithm’s effectiveness in accurately classifying data. On the other hand, the decision tree algorithm demonstrates moderately good performance with an accuracy of 75.03%, an F1-score of 74.77%, a recall of 75.05%, and a precision of 74.73% [2]. Although these metrics suggest reasonable performance, they fall short compared to the neural network algorithm.

Similarly, the random forest algorithm also exhibits decent performance. Still, it is slightly lower than the decision tree algorithm, with an accuracy of 73.43%, an F1-score of 73.56%, a recall of 73.85%, and a precision of 73.67% [2]. Overall, the results highlight the neural network algorithm as the most effective and reliable choice among the three algorithms. However, it is essential to conduct a comprehensive analysis considering additional metrics such as time consumption, loss function, and error rate to understand the algorithms’ performance fully. Based on these comparisons, the proposed model demonstrates superior performance in accuracy, F1-score, recall, precision, time consumption, loss function, and error rate. However, it is important to note that the specific context and requirements of the problem should also be considered when selecting the most suitable model for a given task.

In Table 11, showing the proposed model, as well as the neural network, decision tree, and random forest algorithms, it is evident that each model performs differently across the various evaluation measures. The proposed model’s mean absolute error (MAE) is 0.0163923 [3]. In comparison, the neural network achieves a slightly higher MAE of 0.0236756 [2]. The decision tree model yields a MAE of 0.0174911, and the random forest model obtains a similar MAE of 0.0226886 [1]. Moving on to the mean squared error (MSE), the proposed model achieves the lowest value at 0.00131245, followed by the neural network with 0.00149826. The decision tree model has a MSE of 0.0026259, and the random forest model performs slightly better with a MSE of 0.0017358. The root mean squared error (RMSE) for the proposed model is 0.03499126, and the neural network and decision tree models have RMSE values of 0.03649006 and 0.0503219, respectively. The random forest model performs the best in terms of RMSE, achieving a value of 0.0413318.

When considering the coefficient of determination (R-squared), which measures the proportion of the variance in the dependent variable that the independent variables can explain, the random forest model outperforms the other models with an R-squared value of 0.9886125. The proposed model and neural network follow closely with R-squared values of 0.97114391 and 0.97315592, respectively. The decision tree model achieves an R-squared value of 0.9786136.

The relative absolute error (RAE), which assesses the average difference between predicted and actual values relative to the mean of the actual values, is the lowest for the proposed model at 0.0868276. The neural network model has a higher RAE of 0.1258386, and the decision tree and random forest models yield RAE values of 0.0978889 and 0.0977898, respectively. Lastly, the relative root squared error (RRSE), which measures the average difference between predicted and actual values relative to the range of the actual values, is the lowest for the proposed model at 0.1613141. The neural network, decision tree, and random forest models achieve RRSE values of 0.1769975, 0.1713163, and 0.1624251, respectively.

Finally, based on the provided metrics, the proposed model showcases the best performance in terms of MAE and MSE. However, the random forest model outperforms the other models regarding R-squared, indicating a higher proportion of explained variance. Additionally, the proposed model demonstrates lower RAE and RRSE values, suggesting better accuracy than the other models.

In Table 12, showing the neural network, decision tree, random forest, and proposed models, we can observe variations in the mortality rate, vaccination rate, and other relevant features. The neural network model demonstrates a mortality rate of 0.05 and a vaccination rate of 0.02, and other relevant features are at 0.7. The decision tree model, on the other hand, shows a lower mortality rate of 0.02 and a vaccination rate of 0.01, with other relevant features at 0.85. Comparatively, the random forest model has a higher mortality rate of 0.1 and a vaccination rate of 0.03, and other relevant features are at 0.5. The proposed model falls in between, with a mortality rate of 0.056, a vaccination rate of 0.04, and other applicable features at 0.78.

From these results, it can be inferred that the decision tree model has the lowest mortality rate of 0.02, followed by the proposed model with a mortality rate of 0.056. The neural network and random forest models demonstrate higher mortality rates of 0.05 and 0.1, respectively. Regarding vaccination rates, the decision tree model has the lowest value at 0.01, and the random forest model has the highest value at 0.03. The proposed model falls in between, with a vaccination rate of 0.04. When considering other relevant features, the decision tree model has the highest value at 0.85, followed by the proposed model with 0.78. The neural network model has other relevant features at 0.7, and the random forest model has the lowest value at 0.5.

At last, based on the provided data, the decision tree model showcases the lowest mortality rate and the highest vaccination rate among the algorithms. However, the proposed model performs reasonably well, with a mortality rate higher than the decision tree model but lower than the neural network and random forest models. Additionally, the proposed model demonstrates a moderate vaccination rate and comparatively high values for other relevant features.

A widely used approach to evaluating the performance of a machine-learning model is a confusion matrix. This matrix provides a tabulated representation of the predicted results versus the actual outcomes, which allows for a comprehensive examination of the model’s effectiveness and limitations. By analyzing the matrix, one can obtain a more detailed understanding of the model’s strengths and weaknesses, which can help in making informed decisions about improving its performance in Table 13. A statistical analysis of distributions of data involves using various statistical methods to describe and summarize the characteristics of a dataset. The main objective of this analysis is to gain insight into the underlying patterns and trends within the data. These measures provide information about the typical or average values of the dataset in Table 13.

Among the countries most affected by the SARS-CoV-2 pandemic, the top 10 confirmed cases include the United States, India, Brazil, Russia, the United Kingdom, France, Turkey, Italy, Spain, and Germany. Though some nations have seen a decline in new infections and deaths, others struggle with high transmission rates and overwhelmed healthcare systems. In terms of recovery rates, many countries have shown significant improvements thanks to the availability of vaccines and better treatment options. However, disparities still exist based on factors such as age, underlying health conditions, and access to healthcare Figure 4.

A confusion matrix is a table that is used to evaluate the performance of a machine-learning model by comparing predicted and actual outcomes. It provides a useful way to visualize the true positives, true negatives, false positives, and false negatives generated by a model. Neural networks are a machine-learning model inspired by the structure of the human brain. They consist of layers of interconnected nodes that perform mathematical operations on input data, gradually learning to recognize patterns and make predictions. Decision trees are another type of machine-learning model that work by recursively splitting data into smaller subsets based on the most informative features. They create a tree-like structure that can be easily visualized, making them useful for explaining how a model makes decisions. Random forests are an ensemble method that combine multiple decision trees to improve predictive accuracy. Each tree in the forest is built on a random subset of the training data and a random subset of the available features. This helps to reduce overfitting and increase the generalization ability of the model. In Figure 5, death and recoveries over time are exhibited using the metrics of std, mean and count. In Figure 6, Figure 7 and Figure 8, shows the error rate is a metric used to measure the performance of a machine learning model. It quantifies the number of incorrect predictions made by the model. Loss functions play a vital role in machine learning models as they quantify the discrepancy between predicted outputs and the actual values. Computational time is an important consideration when working with machine learning models. The time required for training and inference can vary depending on the complexity of the model, the size of the dataset, and the available computational resources.

To evaluate the performance of a neural network, a decision tree and random forest use a ROC curve; the network is trained on a labeled dataset and then tested on a separate set of labeled instances. The predicted labels and actual labels are compared to calculate the TPR and FPR at different threshold values, and these values are plotted on the ROC curve in Figure 9, Figure 10 and Figure 11. Finally, the ROC curve is a useful tool for evaluating the performance of binary classifiers such as neural networks, decision trees, and random forests. Plotting the TPR and FPR at different threshold values shows how the classifier’s performance changes as the threshold is adjusted, allowing for easy comparison between different classifiers.

The vast and complex world of machine learning requires a deep understanding of various models, performance metrics, and optimization techniques. To improve the performance of machine-learning models, it is essential to experiment with different hyperparameters, regression techniques, and embedding methods to fine-tune them. By carefully optimizing these factors, our machine-learning models can achieve greater accuracy and efficiency.

Figure 9 shows the optimized value as a red line and the ROC points as a blue line. To be regarded as a good model, the ROC curve must be larger than the optimized region. The model has to be improved if the region is smaller than the optimized one.

In Table 14, the AUC (area under the curve) score is a popular metric for evaluating the performance of binary classifiers. In the context of neural networks, decision trees, and random forests, the AUC score measures how well these models can distinguish between positive and negative samples. A higher AUC score indicates better classifier performance: a score of 0.5 indicates random guessing, and a score of 1.0 indicates perfect classification. Neural networks, decision trees, and random forests can all be trained to optimize the AUC score and improve their ability to classify samples correctly. As compared to other models, the proposed model is the best with a marginal difference.

## 6. Conclusions and Future Work

This study focuses on developing a predictive model for SARS-CoV-2 using machine-learning techniques, aiming to enhance prediction accuracy and provide valuable insights into the factors influencing the spread of the virus. The proposed model incorporates stochastic regression to capture the stochastic nature of virus transmission, considering uncertainties in the data. Additionally, feature selection methods are employed to identify the most relevant features contributing to prediction accuracy. The study also explores the use of neighbor embedding and Sammon mapping algorithms to visualize the high-dimensional SARS-CoV-2 data in a lower-dimensional space, facilitating better interpretation and understanding of underlying patterns. Various algorithms, including the SARS-CoV-2 dataset, combined with ADHPSRNESM, are utilized in the analysis, and a new approach called the proposed model is suggested by combining these methodologies. The research findings can contribute to public health efforts by enabling informed decision making for policymakers and healthcare professionals, ultimately aiding in effectively managing and controlling the SARS-CoV-2 pandemic. The research on enhancing accuracy through feature selection and visualization of high-dimensional data in the context of SARS-CoV-2 has substantial implications for predicting and understanding the spread of the virus. By employing various feature selection methods, the study aims to identify the most relevant and informative factors that contribute to the accuracy of SARS-CoV-2 predictions, eliminating noise and unnecessary variables for improved prediction performance. Additionally, the research incorporates neighbor embedding and Sammon mapping algorithms to visualize the high-dimensional SARS-CoV-2 data in a lower-dimensional space, enabling the identification of clusters, trends, and relationships that may not be apparent in the original data.

These findings have practical implications for public health efforts, empowering policymakers and healthcare professionals to allocate resources, implement control measures, and target interventions based on accurate predictions and a better understanding of the underlying patterns driving SARS-CoV-2 transmission. Ultimately, this research contributes to the global response in managing and controlling the SARS-CoV-2 outbreak, minimizing its impact on affected populations.

## Figures and Tables

**Figure 1 bioengineering-10-00880-f001:**
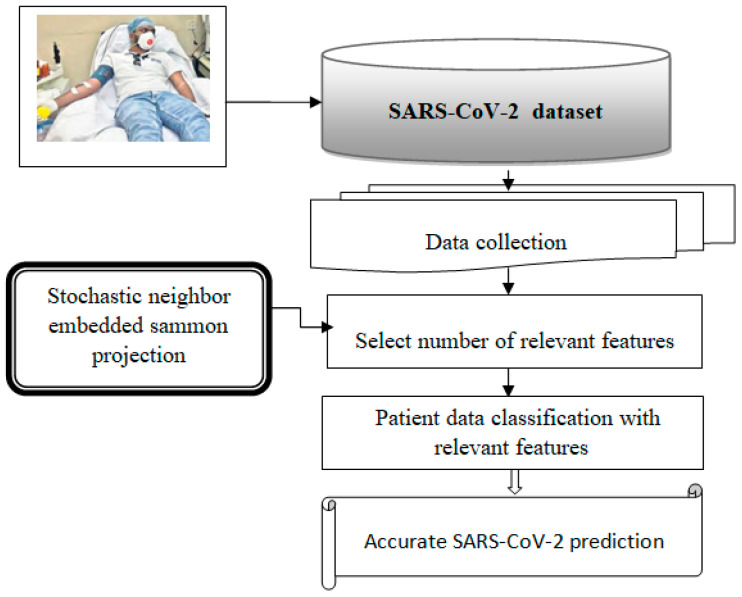
Architecture of the proposed mode.

**Figure 2 bioengineering-10-00880-f002:**
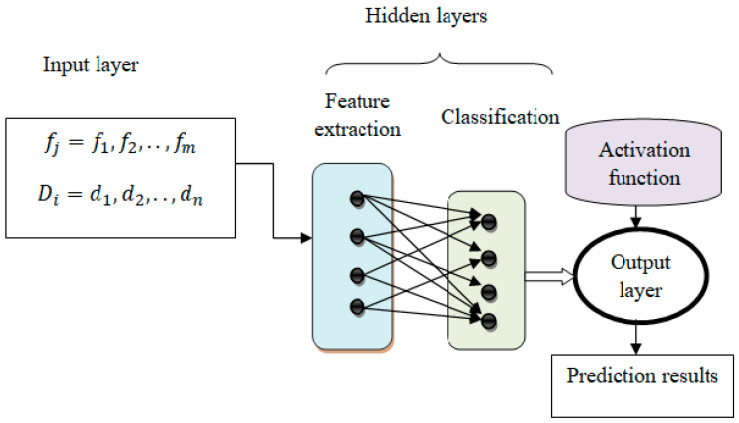
Structure of neural networks with deep connectivity shift-invariant convolution.

**Figure 3 bioengineering-10-00880-f003:**
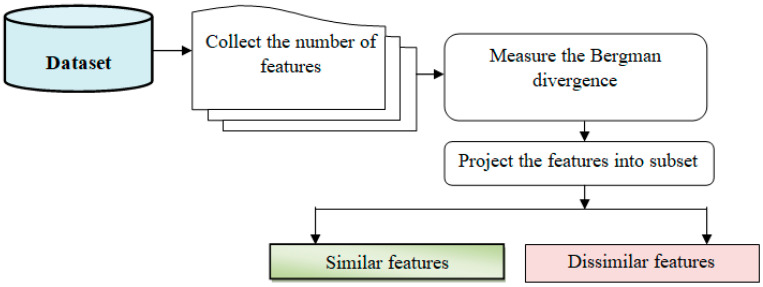
Feature selection procedure block diagram.

**Figure 4 bioengineering-10-00880-f004:**
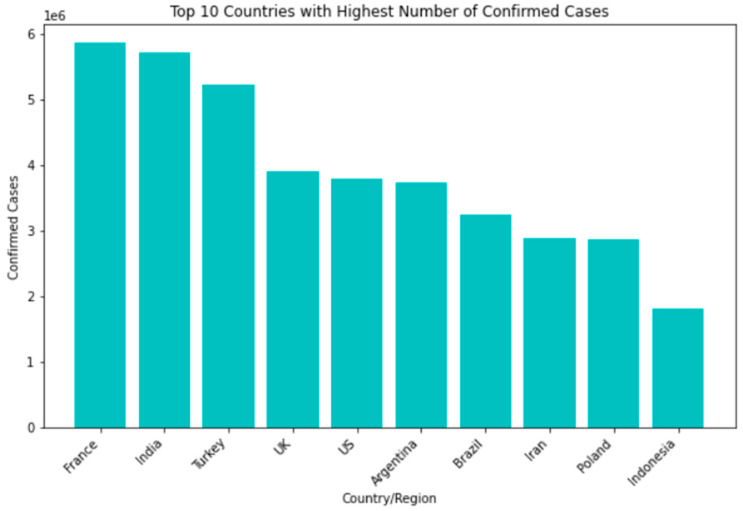
Top 10 countries with highest number of confirmed cases.

**Figure 5 bioengineering-10-00880-f005:**
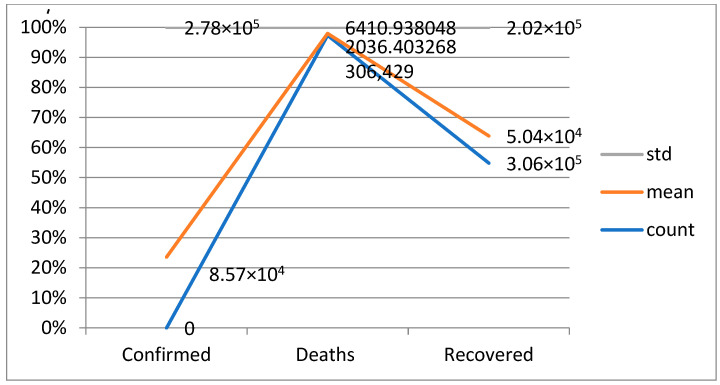
Death and recoveries over time.

**Figure 6 bioengineering-10-00880-f006:**
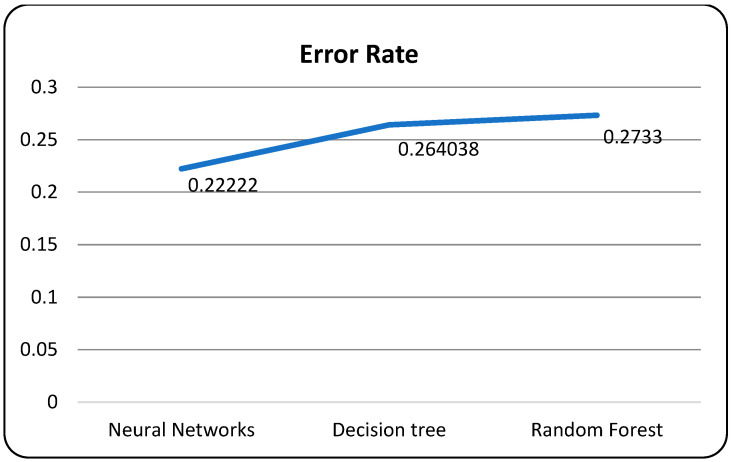
Models with error rate.

**Figure 7 bioengineering-10-00880-f007:**
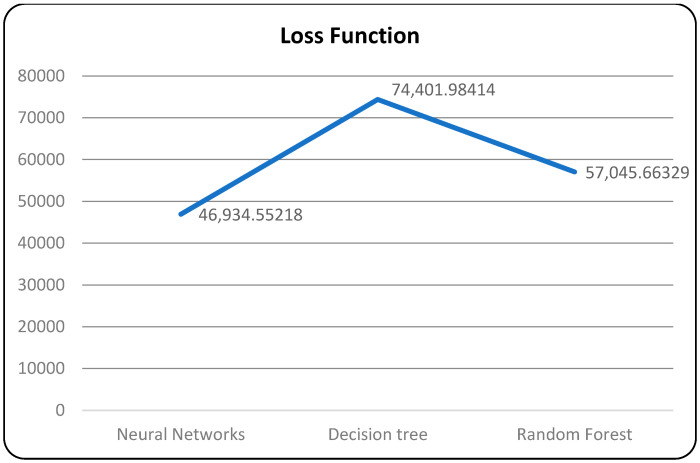
Models with loss function.

**Figure 8 bioengineering-10-00880-f008:**
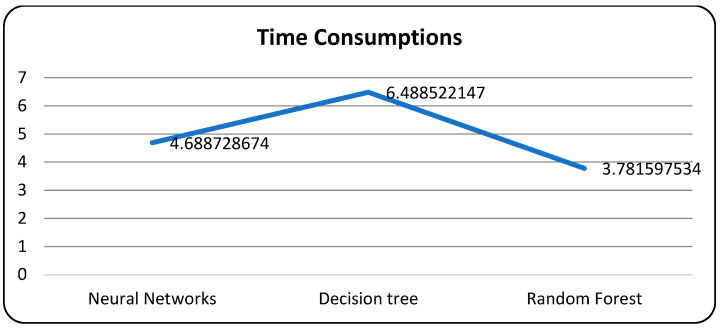
Models with computational time.

**Figure 9 bioengineering-10-00880-f009:**
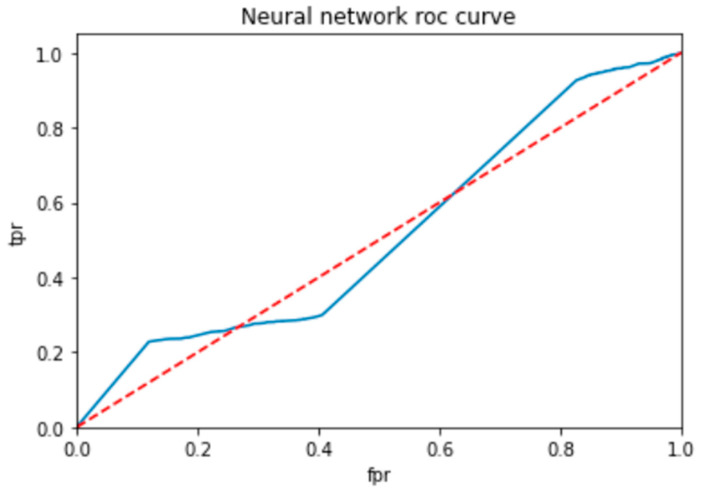
ROC for neural network classifier.

**Figure 10 bioengineering-10-00880-f010:**
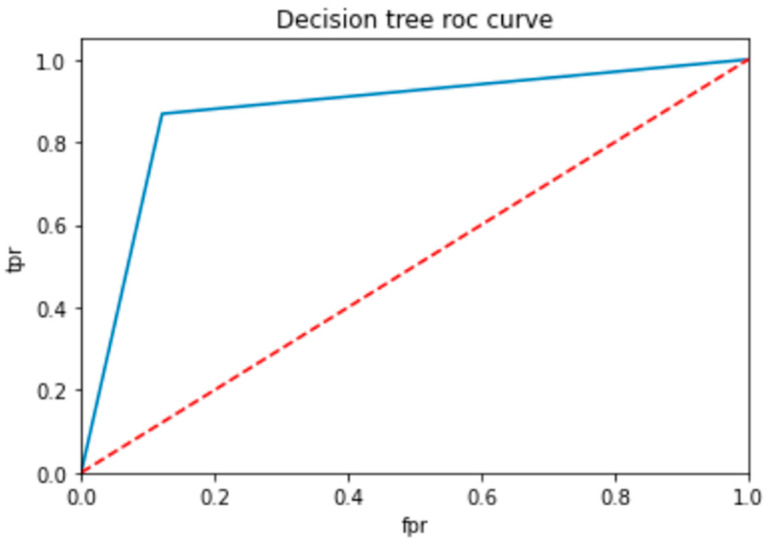
ROC for decision tree classifier.

**Figure 11 bioengineering-10-00880-f011:**
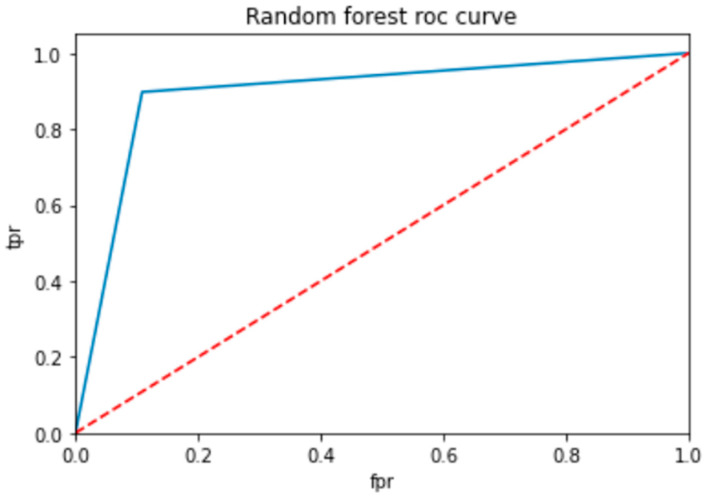
ROC for random forest classifier.

**Table 1 bioengineering-10-00880-t001:** Feature analysis for class 1.

Feature	Age	Symptoms	Travel	Chronic	*Y*	Class 1
Training	>50	37 ℃	>28 days	False	<0.5	Low risk
Testing	>50	37 ℃	>28 days	False		

**Table 2 bioengineering-10-00880-t002:** Feature analysis for class 2.

Feature	Age	Symptoms	Travel	Chronic	R	Class 2
Training	>50	37 ℃<T<39 ℃	14–28 days	True	0.5	Medium risk
Testing	>50	37 ℃<T<39 ℃	14–28 days	True		

**Table 3 bioengineering-10-00880-t003:** Feature analysis for class 3.

Feature	Age	Symptoms	Travel	Chronic	R	Class 3
Training	>50	>39 ℃	<14 days	True	>0.5	High risk
Testing	>50	>39 ℃	<14 days	True		

**Table 4 bioengineering-10-00880-t004:** Comparison with other models.

Patient Data	Prediction Accuracy (%)		
	Proposed Model	Deep-LSTM	MTGP
1000	89	85	82
2000	91	88	84
3000	90	87	83
4000	91	88	85
5000	90	87	84
6000	92	88	85
7000	91	87	84
8000	90	86	83
9000	91	87	84
10,000	90	86	83

**Table 5 bioengineering-10-00880-t005:** Comparison of the FPR.

Patient Data	FPR (%)		
	Proposed Model	Deep-LSTM	MTGP
1000	11	15	18
2000	9	12	16
3000	10	13	17
4000	9	12	15
5000	10	13	16
6000	8	12	15
7000	9	13	16
8000	10	14	17
9000	9	13	16
10,000	10	14	17

**Table 6 bioengineering-10-00880-t006:** Comparison of prediction time.

Patient Size	Prediction Time (in ms)		
	Proposed Model	Deep-LSTM	MTGP
1000	23	26	28
2000	27	32	36
3000	33	36	39
4000	37	40	42
5000	40	45	48
6000	45	48	51
7000	48	53	56
8000	52	56	60
9000	56	59	63
10,000	58	60	65

**Table 7 bioengineering-10-00880-t007:** Comparison of space complexity.

Patient Data	Space Complexity (MB)		
	Proposed Model	Deep-LSTM	MTGP
1000	21	24	27
2000	24	26	30
3000	27	30	33
4000	32	36	40
5000	35	40	45
6000	39	43	48
7000	43	46	50
8000	46	50	52
9000	51	54	57
10,000	55	58	61

**Table 8 bioengineering-10-00880-t008:** Machine-learning techniques with core analysis.

Algorithm	Accuracy	F1-Score	Recall	Precision	Time Consumption	Loss Function	Error Rate
Neural network	97.78%	98.80%	98.32%	98.28%	4.688728674	46,934.55218	0.02222
Decision tree	73.59%	73.44%	73.59%	73.60%	6.488522147	74,401.98414	0.2641
Random forest	72.67%	72.15%	72.66%	72.68%	3.781597534	57,045.66329	0.2733

**Table 9 bioengineering-10-00880-t009:** Dataset with Adam optimizer.

Algorithm	Accuracy	F1-Score	Recall	Precision	Time Consumption	Loss Function	Error Rate
Neural network	98.78%	99.10%	98.92%	98.92%	3.278726674	34,934.45318	0.0122
Decision tree	74.59%	74.56%	74.64%	74.42%	5.688512147	64,321.88314	0.2541
Random forest	72.99%	73.35%	73.44%	73.36%	2.582577534	48,035.56229	0.2701
Proposed model	99.03%	99.32%	98.98%	99.06%	2.12368745	33,876.36407	0.0118

**Table 10 bioengineering-10-00880-t010:** Dataset with Adam optimizer and hyperparameters.

Algorithm	Accuracy	F1-Score	Recall	Precision	Time Consumption	Loss Function	Error Rate
Neural network	99.22%	99.31%	99.33%	99.23%	2.818727	34,933.2532	0.78
Decision tree	75.03%	74.77%	75.05%	74.73%	5.228512	64,320.6831	0.2497
Random forest	73.43%	73.56%	73.85%	73.67%	2.122578	48,034.3623	0.2701
Proposed model	99.41%	99.46%	99.45%	99.37%	2.032367	33,842.2328	0.2391

**Table 11 bioengineering-10-00880-t011:** Dataset with Adam optimizer, hyperparameters with stochastic regression.

Metrics	Proposed Model	Neural Network	Decision Tree	Random Forest
MAE	0.0163923	0.0236756	0.0174911	0.0226886
MSE	0.00131245	0.00149826	0.0026259	0.0017358
RME	0.03499126	0.03649006	0.0503219	0.0413318
R-Squared	0.97114391	0.97315592	0.9786136	0.9886125
RAE	0.0868276	0.1258386	0.0978889	0.0977898
RRSE	0.1613141	0.1769975	0.1713163	0.1624251

**Table 12 bioengineering-10-00880-t012:** Dataset with Adam optimizer, hyperparameters with stochastic regression with Sammon mapping.

Algorithm	Mortality Rate in %	Vaccination Rate in %	Other Relevant Features in %
Neural Network	0.05	0.02	0.7
Decision Tree	0.02	0.01	0.85
Random Forest	0.1	0.03	0.5
Proposed Model	0.056	0.04	0.78

**Table 13 bioengineering-10-00880-t013:** Distributions of the data.

Statistical Measure	Confirmed	Deaths	Recovered
count	3.06 × 10^3^	306,429	3.06 × 10^5^
mean	8.567091 × 10^4^	2036.403268	5.04 × 10^4^
std	2.78 × 10^5^	6410.938048	2.02 × 10^5^
min	−3.03 × 10^5^	−178.000000	−8.54 × 10^5^
25%	1.04 × 10^3^	13	1.10 × 10^1^
50%	1.037500 × 10^4^	192	1.75 × 10^3^
75%	5.08 × 10^4^	1322	2.03 × 10^4^
max	5.86 × 10^6^	112,385	6.40 × 10^6^

**Table 14 bioengineering-10-00880-t014:** AUC scores for classifiers.

Classifiers	AUC Score
Neural Network	0.874033334
Decision Tree	0.873343619
Random Forest	0.894642082
Propose Model	0.879465732

## Data Availability

https://www.kaggle.com/datasets/plameneduardo/sarscov2-ctscan-dataset, accessed on 24 May 2023.

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
