# Peer review of "Evaluation of Mutual Information and Feature Selection for SARS-CoV-2 Respiratory Infection"

_bioengineering, 2023, doi:10.3390/bioengineering10070880_

Round 1
Reviewer 1 Report
First of all, the goal of the research should be stated in a precise and clear way. Predicting SARS-CoV-2 is a very vague and generic concept. The extremely long and dispersive introduction does not help to focus on the goal of the research.
The draft must be radically revised: 1) creating a short (1-2 pages) and clear introduction with the goal expressed in a precise, technical and clear way; 2) revise language and punctuation; 3) move many of the considerations added to the introduction into the discussion, that still has to be concise (3-4 pages); 4) clear considerations of interpretability and explainability of the approach must be added; 5) clear considerations about the risks of bias and the de-biasing methos used must be expressed; 6) tables and figures are too many, often of very scarce interest considering the goal of the research, and often lack clear explanations. Remove many of them and, only if deemed necessary, add some of them as supplementary files; 7) clearly state the field of application and the limitations (also ethical limitations related to interpretability and explainability); 8) 2-3 lines at the very beginning of the manuscript regarding the main characteristics of interest of the virus must be added. For example, the way of contagion (stressing its persistance on vital and non-vital surfaces), the higher risks for hospitalized rather than non-hospitalized patients etc. If the authors need some references to evaluate this crucial characteristics: doi: 10.3389/fcell.2020.00618; doi: 10.1007/s00414-021-02753-2. ; https://doi.org/10.1186/s12879-021-06222-4
language and in particular punctuation must be revised
Reviewer 2 Report
Please revise the title to make it shorter while highlighting the novelty of the machine learning approaches.
Remove the heading of subsection 1.1 as it is unusual to have only one subsection in a section.
To improve the introduction, incorporate statistical information regarding COVID-19.
Update the literature review with recent articles. Consider including the following references:
-
Supply chain disruption during the COVID-19 pandemic: Recognizing potential disruption management strategies. International Journal of Disaster Risk Reduction, 102983, 2022.
-
Sustainable and robust home healthcare logistics: A response to the COVID-19 pandemic. Symmetry, 14(2), 193, 2022.
Please include more recent works in the literature review section, focusing on the area of COVID-19.
Merge subsection 2.1 with Section 2, as it is unnecessary to have a separate subsection.
Figure 9 requires improvement in resolution to be publishable. Please enhance it accordingly.
Similarly, Figure 10 and 11 are of poor quality and need improvement.
Consider removing excess tables and figures since there are many of them.
Ensure that all abbreviations are defined in their first appearance in the text.
Revise the English writing and presentation throughout the paper. For instance, the conclusion should be divided into paragraphs. Begin by summarizing the paper and highlighting the main findings. Then, discuss the limitations, managerial insights, and propose future works.
Revise the English writing and presentation throughout the paper. For instance, the conclusion should be divided into paragraphs. Begin by summarizing the paper and highlighting the main findings. Then, discuss the limitations, managerial insights, and propose future works.
Reviewer 3 Report
The study conducted by Sekar Kidambi Raju et al. aims to develop a predictive model for SARS-CoV-2 using machine learning techniques and explore various feature selection methods to enhance the accuracy of predictions. The global impact of the SARS-CoV-2 pandemic has emphasized the need for accurate prediction of its spread to facilitate effective planning and resource allocation.
In this research, stochastic regression is employed to capture the stochastic nature of virus transmission and consider uncertainties in the data. Feature selection techniques are used to identify the most relevant and informative features that contribute to the prediction accuracy. Additionally, neighbor embedding and Sammon mapping algorithms are explored to visualize the high-dimensional SARS-CoV-2 data in a lower-dimensional space, enabling better interpretation and understanding of underlying patterns.
The study focuses on machine learning techniques for predicting SARS-CoV-2 infections and utilizes statistical measures in healthcare, such as confirmed cases, deaths, and recoveries. Country-wise dynamics of the pandemic are analyzed using machine learning models. The performance of various algorithms, including neural networks, decision trees, random forests, adam optimizer, hyperparameters, stochastic regression, neighbor embedding, and Sammon mapping, is assessed.
By combining the pre-processed and feature-extracted SARS-CoV-2 dataset with ADHPSRNESM, a new orchestration model is proposed to improve prediction precision and accuracy. The incorporation of these techniques aims to enhance the accuracy of SARS-CoV-2 predictions and provide valuable insights into the factors influencing the virus's spread.
The findings of this research can contribute to public health efforts by enabling policymakers and healthcare professionals to make informed decisions based on accurate predictions and a better understanding of the pandemic's dynamics.
Novel Feature Selection Techniques: The study introduces feature selection algorithms specifically tailored to SARS-CoV-2 data, taking into account the unique characteristics and dynamics of the virus. Traditional feature selection methods may not effectively capture the intricate relationships among variables in complex datasets like SARS-CoV-2. Therefore, the proposed novel feature selection techniques aim to identify the most relevant predictors for accurate predictions.
Stochastic Regression Framework: The research incorporates a stochastic regression framework that utilizes probabilistic modeling techniques to account for the inherent randomness and unpredictability of the pandemic. By leveraging stochastic regression, the approach aims to provide more robust and reliable predictions, enabling better decision-making in uncertain situations.
Neighbour Embedding and Sammon Mapping: The study utilizes neighbor embedding algorithms and Sammon mapping, which are nonlinear dimensionality reduction techniques. These techniques reduce the dimensionality of the SARS-CoV-2 data while preserving local relationships between instances and emphasizing the preservation of pairwise distances. By leveraging these techniques, the approach aims to capture intricate relationships and similarities between SARS-CoV-2 cases, leading to improved prediction accuracy and interpretability.
Advanced Machine Learning Models: The research employs advanced machine learning models, including ensemble methods and hybrid models, to enhance the predictive capabilities of the system. By utilizing these models, the study aims to generate accurate and reliable forecasts for various outcomes related to SARS-CoV-2 spread and severity.
Overall, the combination of these novel elements in the research work suggests that it introduces innovative approaches for SARS-CoV-2 prediction and analysis. However, a comprehensive evaluation of the research paper would require a thorough examination of the methodology, experimental setup, and results provided in the full article.
It is recommended to edit the table and adjust it according to the provided template in order to enhance readability and adhere to standard formatting conventions. Incorporating the template guidelines will help create a visually cohesive and standardized appearance for the table, potentially utilizing the color orange for better visual differentiation and consistency.
Considering the content provided, it may be worth evaluating the necessity of all the included text and considering the possibility of moving certain parts in supplementary materials. This evaluation aims to streamline the text and improve its overall clarity and conciseness. By selectively removing or relocating non-essential information, the focus can be directed towards the most relevant and crucial aspects, enhancing the effectiveness of the communication.
To improve the presentation of the figures, it is recommended to recreate them, potentially combining multiple subpanels into a single figure. This approach can enhance the overall visual coherence and organization of the figures, making them more concise and easier to interpret. By consolidating the information into a single figure with subpanels, the clarity and effectiveness of the visual representation can be enhanced.
It is worth noting that the provided text does not include a specific section on limitations, which is an essential component of a research study. A limitations section is crucial as it acknowledges the potential constraints, shortcomings, and boundaries of the research work. This section helps to provide a balanced perspective on the study's findings and interpretations, ensuring transparency and highlighting areas where further research is needed. Therefore, it is recommended to include a dedicated limitations section to address any potential limitations or constraints that may have impacted the research methodology, data analysis, or generalizability of the results.
see above
Round 2
Reviewer 2 Report
The authors have revised the paper in good order. All my comments have been addressed successfully. I agree with the publication of this paper.
Reviewer 3 Report
can be published